# Advances in Oral Drug Delivery Systems: Challenges and Opportunities

**DOI:** 10.3390/pharmaceutics15020484

**Published:** 2023-02-01

**Authors:** Jie Lou, Hongli Duan, Qin Qin, Zhipeng Teng, Fengxu Gan, Xiaofang Zhou, Xing Zhou

**Affiliations:** 1School of Pharmacy and Bioengineering, Chongqing University of Technology, Chongqing 400054, China; 2Chongqing Key Laboratory of Medicinal Chemistry & Molecular Pharmacology, Chongqing University of Technology, Chongqing 400054, China; 3Department of Neurosurgery, Chongqing Hospital of Traditional Chinese Medicine, Chongqing 400011, China; 4Department of Pharmacy, Chongqing Hospital of Traditional Chinese Medicine, Chongqing 400011, China

**Keywords:** oral drug delivery system, gastrointestinal barriers, oral target delivery, oral nanomedicine, novel drug delivery system

## Abstract

The oral route is the most preferred route for systemic and local drug delivery. However, the oral drug delivery system faces the harsh physiological and physicochemical environment of the gastrointestinal tract, which limits the bioavailability and targeted design of oral drug delivery system. Innovative pharmaceutical approaches including nanoparticulate formulations, biomimetic drug formulations, and microfabricated devices have been explored to optimize drug targeting and bioavailability. In this review, the anatomical factors, biochemical factors, and physiology factors that influence delivering drug via oral route are discussed and recent advance in conventional and novel oral drug delivery approaches for improving drug bioavailability and targeting ability are highlighted. We also address the challenges and opportunities of oral drug delivery systems in future.

## 1. Introduction

Oral administration is the most often used treatment for both systemic and local gastrointestinal diseases [1,2]. Despite the apparent advantages, oral drug delivery remains challenging due to the harsh gastrointestinal tract (GIT) microenvironment and a number of physiological barriers, including gastrointestinal anatomy factors, biochemistry factors, and physiology factors. The different parts of the GIT, including mouth cavity, esophagus, stomach, small intestine, and colon, play an important role in the digestion of food and absorption of medicine. The different anatomical characteristics, such as limited surface in oral cavity, gastric mucin–bicarbonate barrier, and enteral enzymes could also be obstacles for drug absorption. Considerable efforts have been made to overcome these issues, which are mainly based on improved comprehension of the healthy and diseased physiology characters of the GIT. Conventional drug delivery systems including normal tablets, capsules, or sterile drug preparations, are associated with limitations, including low site-specific accumulation of drugs, unfavorable body distribution, adverse side effects, etc. [3]. Therefore, the development of novel localized and systematic targeted drug delivery systems is urgent. The application of nanomedicines and novel drug delivery devices were considered as the most promising innovative pharmaceutical approaches in oral drug delivery systems [4].

Since oral delivery has very limited oral bioavailability, nanoparticles have presented great potential in drug delivery [5]. Both organic and inorganic nanoparticles [6] have been studied to improve tolerability, pharmacologic specificity, biodegradability, and targeting for oral drugs [7,8]. Numerous nanocarriers including nanoparticles [9,10], liposomes [11,12], emulsions [13] etc., have been applied for oral drugs. Most nanocarriers showed advantages in protecting drugs from harsh conditions in the GIT, increasing the absorption into the circulatory system from GIT, targeting specific sites and guaranteeing a controlled release.

Novel microfabricated devices showed great potential to break the barrier in GIT thereby improving oral bioavailability of drugs [14]. Microfabricated oral drug delivery systems, mainly including patch-like structures, microcontainers, and microwells, are speculated to exert a unidirectional drug release and promote oriented adhesion to the intestine wall, and showed a protective effect for drugs to their final destination of delivery. Due to mucus attachment of penetration, the unidirectional release and the sustained resident time will allow more drug to cross the gastrointestinal wall and be absorbed, resulting in an improvement of oral bioavailability [15].

Most oral drug delivery systems also focused on targeting local gastrointestinal diseases, such as gastric diseases [16], oral carcinoma [17], inflammatory bowel disease (IBD) [18], and colon cancer [11], but with rapid development of pharmaceutical technology, materials science, and physiological study of diseases, tremendous advances have been made to develop oral targeted nanoparticle preparations that can target drugs to focal sites outside the GIT [19]. In this review, the biological factors that affect the oral administration and applications of nanomedicines and microfabricated devices in and beyond GIT are summarized.

## 2. Biological Barriers to Oral Drug Delivery Systems

On account of its easy application, being painless, low expense, wide drug assimilation/distribution, and high patient compliance, the oral route is the most common way for patients. However, the efficiency of many oral medicines is still limited due to various physiological barriers, resulting in low permeability, and drug degradation [20]. The limitations of oral drug delivery can be summarized by anatomy factors, biochemistry factors, and physiology factors in GIT.

### 2.1. Anatomical Factors

Anatomically, the GIT consists of the oral cavity, esophagus, stomach, small intestine, and colon, each part having different factors that affect drug delivery [21]. The different anatomical characteristics of GIT show varying effects on drug absorption.

The oral cavity is covered by oral mucosa, it has a mild microenvironment, easy accessibility, facile access to circulation, good permeability, and absorption of drugs [20,22]. However, the limited surface of oral cavity, saliva, and enzymatic composition are the main barriers of drug delivery in mouth [20].

Due to the low permeability and short residence time of drugs, the esophagus is not a prime target for drug delivery [23,24].

The stomach exhibits a strong acid environment with a pH range of 1.0–2.5, which can break down food, ectogenic pathogens [25], and acid-labile drugs [26], which makes it the harshest barrier to drug absorption. In addition, the stomach possesses extrinsic epithelial cells [27] and a mucin–bicarbonate barrier [28]. The tight junctions beneath the intrinsic barrier also limit the drug absorption. Moreover, pepsins in the stomach can lead to the inactivation of protein drugs.

The small intestine has a huge surface area due to the villi and microvilli in the intestinal lumen [29,30]. The small intestine is regarded as a prime site for oral drug delivery due to the tremendous surface and diverse transport routes. The intestinal mucosa can recognize and transfer ectogenic antigens to the immune system [31,32]. However, there are still some challenges of small intestine drug delivery arising from its special physiology. The harsh stomach chemical microenvironment, pancreatic enzymes, bile salts, and the mucosal layer decrease the drug bioavailability. Drug delivery systems that can increase their retention time at villi and microvilli, improve lipid solubility, and interact with specific receptor or carrier are able to increase their overall bioavailability.

The colon exhibits a higher pH environment and much longer residence time compared with the upper GIT, and the enzyme activity in colon is relatively low [33,34]. Moreover, drugs can be metabolized by the gut microflora which effects the release characters of drugs. Targeting drugs to the colon is of great significance for treating bowel diseases with fewer side effects and lower drug dosage. However, the inherent difference of the gastric emptying time and microflora in different people remains a major drawback for colon targeting [35].

### 2.2. Biochemical Factors

Different pH environments and digestive enzymes were regarded as the main biochemical barriers for oral drug delivery systems. The pH varies distinctly in different parts of the GIT, it rises gradually from the stomach to the colon in the range from 1 to 8 [36,37]. The variation from acidic to alkaline environment affects the drugs’ activities and bioavailability. pH variation not only affects drug delivery, but also a route for targeting the design of oral drugs.

The existence of various enzymes will critically influence the bioavailability of drugs in the GIT, especially for protein drugs. There are over 400 different species of aerobic and anaerobic microorganisms in the colon; they can produce hydrolytic and reductive metabolizing enzymes, which can catalyze the metabolism of xenobiotics and other biomolecules. Polysaccharides can only be metabolized in the colon by anaerobic bacteria and be stable in the stomach and intestine, making it possible for colon-targeted drug delivery.

Since drugs are also susceptible to colonic enzymes and generate biotransformation, the “prodrug” approach is often used for the colon-specific drug delivery [33].

The main anatomical and biochemical barriers of the oral administration are summarized in Table 1.

### 2.3. Physiology Factors

The GIT exerts a low permeability to the bloodstream and extraneous substances, which restricts the bioavailability and absorption of drugs. The physiological barriers mainly consist of epithelium cellular barrier and the mucus barrier.

The gastrointestinal epithelium is a phospholipid bilayer membrane, which allows the penetration and absorption of lipophilic macromolecules [38], while it is a primary absorption barrier for hydrophilicity and macromolecules [39]. The existence of tight junctions between adjacent cells also limits the paracellular pathway for hydrophilic drug [40].

Mucus is a dynamic semipermeable barrier, which restricts the direct interaction of drugs with epithelial cells [26]. Mucus is a viscous gel formed by mucins and glycoproteins; it can serve as a lubricant for ingested food and also a strong barrier to entrap foreign particles and eliminate potentially harmful compounds and bacteria [41,42,43,44,45]. Secreted mucins are linked together through disulfide bonds to form highly glycosylated macromolecules, which makes the mucin complex more stable and protects them from enzymatic degradation [28]. The mucus structure and intermolecular interactions dictate the permeation of peptides, large molecules, and microorganisms through the mucus layer [46,47].

## 3. Application of the Oral Drug Delivery Systems

### 3.1. Local Targeting to GIT

Based on the physiological characteristics of different GIT sites, oral drug delivery systems are commonly used to deliver drugs specifically targeting the GIT. The main local targeting delivery strategies are summarized in Figure 1.

#### 3.1.1. Gastroretentive Drug Delivery Systems

The primary strategy of stomach targeting is to prolong the gastric residence time of the formulations. The strategy is suitable for drugs that are mainly absorbed or work in the stomach or upper GIT, or for drugs that are not stable in intestinal alkaline conditions [48,49]. The gastroretentive delivery systems can also be used for sustained or controlled drug release, which is beneficial for reducing the fluctuations of systemic drug concentrations, decreasing the frequency of administration and increase patient compliance to drugs [50]. Although a large number of gastroretentive formulations were studied, there is little progress on the clinical translation. The most frequently used gastric retention preparations are gastric floating, gastric expandable, and mucoadhesive and high-density formulations [51,52,53,54,55,56,57,58].

The gastric residence time can also be prolonged by gastric floating drug delivery system, which is suitable for treating the local infections in the GIT. Serdar et al. reported a pramipexole loaded self-inflating effervescence-based electrospun nanofibers [59]. The formulation is embedded in cast films, which are composed of polyethylene oxide/sodium bicarbonate. The sodium bicarbonate can generate carbon dioxide gas when it contacts acidic gastric fluid, the nanofibers entrap the gas bubbles into the swollen network and results in floating in the stomach. The formulation is able to float when it enters gastric fluid and remain unsinkable even after 24 h.

The folding design is often used to prepare the gastric expandable systems [60]. The systems composed of elastic or collapsible materials are folded and put into a small capsule; when in contact with gastric fluid, they can expand to dimensions greater than the pyloric sphincter, resulting in retention in the stomach. Issam et al. prepared a gabapentin-loaded expandable gastroretentive controlled formulation [61]. Ethanol, different types of Eudragit L, gelatin, gabapentin, and poloxamer P407 were used as matrix and were made into dried layers. The resultant layer was processed into accordion shape and placed in a “00”-sized gelatin capsule. This expandable formulation was able to unfold in less than 15 min after entering stomach and exerted a controlled release in 6 h [61].

Mangesh R. Bhalekar et al. developed a simvastatin-loaded mucoadhesive gastric mucoadhesive drug delivery system using thiomers to enhance the gastric retentive effects [62]. The xyloglucan was used as the main matrix, it was chemically modified to carboxymethyl derivative, conjugated with cysteine, and converted to thiomer. The thiomer tablets containing simvastatin demonstrated increased mucoadhesion force, drug release retardation, and in vitro permeation, and the results were proportional to the increase of thiomer amount. The in vivo study showed tablets that using thiomer had better drug permeation and stronger mucoadhesive force.

Ndidi et al. prepared a gastroretentive levodopa-loaded drug delivery system using inter polymeric blend (IPB) nano-enabled strategy [52]. The IPB system was mesoporous with an average density of 1.41 mg/mm^3^ and rigidity of 75.05 N/mm. The IPB nano-enabled gastroretentive levodopa-loaded drug delivery system exhibited a prolonged period of drug release.

Ankush et al. prepared zero valent iron nanoparticles (ZVINPs) using high-density gastroretentive pellets to improve the bioavailability of iron [53]. The high density gastroretentive ZVINPs pellets increased more than 2-fold oral bioavailability of iron in vivo studies. In acute treatment animals, no evidence of liver damage was observed, while few complications were observed in chronic treatments. The study proved that the oral bioavailability of iron could be increased by ZVINPs pellets; however, the dose toxicity should be noticed during clinical evaluation.

#### 3.1.2. Small Intestine Drug Delivery Systems

Small intestine-targeted drug delivery is usually achieved by gastroretentive, pH-dependent, and mucoadhesive strategies, which can be advantageous for drugs mainly absorbed in small intestine [51,56].

The pH-responsive formulations composed of coatings or matrices that are pH-responsive, which can prevent drugs from degradation induced by gastric enzyme or gastric fluid acidity, and decrease irritation to the GIT mucosa [39,63]. Hydrogels, nanoparticles, microspheres, mini-tablets, etc., composed of natural or synthetic materials are often used as pH-responsive carriers to achieve intestine controlled release [64,65,66,67,68,69,70,71].

The solid enteric-coated dosage forms, including tablets and capsules, are often applied in small intestine targeted drug delivery [72,73]. An enteric-coated formulation is often composed of a polymer coating, which can form a barrier over the surface and allow transit to the small intestine, and the drug cannot be released until it reaches the intestine [73]. However, the drug’s release and formulation disintegration of the enteric coating preparations may be unstable due to the limited dissolution of drugs or degradation of the coating polymers compared to the intestine transit time [72,74]. Moreover, individual variability in the gastrointestinal emptying time, pH, and composition of gastrointestinal fluid will also influence drug release from enteric-coated formulations [75].

Xu et al. developed a biodegradable polymeric carrier to orally deliver a vaccine to small intestine and target the antigen presenting cells in Peyer’s patches region. The protein vaccine, bovine serum albumin, was enwrapped into the mannosylated chitosan nanoparticles (MCS NPs). MCS NPs were coated with Eudragit(R) L100 and were found to be specifically accumulated in Peyer’s patches sites in GIT [76].

In order to improve the absorption of drugs, with the main strategy of extending the contact time with intestinal mucosa, mucoadhesive formulations have been investigated [77,78,79,80]. Drug release from formulation is influenced by different kinds of formulation factors, including polymer ingredients, mucosal adhesive strength, drug concentration, etc. [81]. Intestinal patches are most often used mucoadhesive formulations; they are usually composed of three layers, including a certain pH sensitive layer, a mucoadhesive drug reservoir layer, and a backing layer. Amrita et al. prepared insulin oral-delivered intestinal mucoadhesive devices [77]. The devices were composed of sodium carboxymethyl cellulose, ethyl cellulose, Eudragit^®^ E PO, and pectin; after being pressed into patches, they were placed in capsules coated with Eudragit^®^ L100 to survive the acid environment of stomach. The devices could release the loaded drugs unidirectionally after adhering to intestinal mucosa and avoid gut enzymatic degradation. The devices were proven to have good mucoadhesive ability to porcine intestine and could completely release drugs within 3–4 h in animal studies.

#### 3.1.3. Colon Targeting

Colon targeting drug delivery is an often-used approach to cure the local colon diseases; they can exert advantages of rapid action, reduce drug dosage, and minimize harmful side effects [82].

##### Prodrugs

The prodrug strategy was often used to achieve colon targeting. Prodrugs are usually obtained by covalently bonding the drug moiety with another chemical substance [83]. Prodrugs are designed to be barely absorbed and hydrolyzed in the upper GIT, and the active drug can be released in colon due to the enzymatic or pH hydrolysis [84].

5-aminosalicylic acid (5-ASA) is a frequently used anti-inflammation drug; it shows an excellent effect in treating colitis and inflammatory bowel disease.

Drug moieties such as sulfapyridine and 5-ASA have been used to prepare prodrugs of sulfasalazine and olsalazine. The azoreductase produced by colonic bacteria can digest the azo-linkage and thus 5-ASA can be released. The therapeutic effects of the above drugs were studied and the importance of cleavage rate of the azo bond was proven by Sousa et al. [85]. The prodrugs’ metabolism rates were determined in the environment of colonic bacteria, and results were relevant to those in humans [85].

Soojin Kim et al. developed an endoplasmic reticulum stress attenuator that was used to treat colitis. The study used 4-phenylbutyric acid (4-PBA) as model drug, then yielded 4-PBA-glutamic acid (PBA-GA) and 4-PBA-aspartic acid (PBA-AA) conjugates by conjugation with acidic amino acids. The derivatives can be converted to 4-PBA in colon and PBA-GA converted more than PBA-AA. In vivo studies found PBA-GA could alleviate the colon inflammation colitis rats, which in all proved that PBA-GA could be colon-targeted and effective against colitis in rats [86].

##### Enzyme Response Approach

To achieve colon target by enzyme response, polymeric systems were developed using degradable polymers which were sensitive to specific colon enzymes. The strategy can also be used in prodrugs; they transformed to active forms when contacted with specific colonic enzymes with rupture of sensitive polymers.

Chen et al. [87] obtained resistant starch film-coated 5-ASA microparticles using the extrusion–spheronization method. The microparticles were fluorescein-labeled and orally administrated, and they were found to be highly acidic and enzyme-resistible in upper GIT, indicating that the formulation had good colon-specificity [82].

Some polysaccharides such as pectin, chitosan, or inulin were used to develop colonic targeted formulations due to the characteristic of being degraded by colon enzymes. Novel and improved proposals of the materials have been exploited. Recently, Günter et al. prepared prednisolone-loaded calcium pectinate gel beads with low methyl-esterified pectins [88]. Drug release in simulated GIT media was studied, the prednisolone was found to be mainly released in simulated colonic medium, which proved that the beads might be an effective approach for colon-targeting delivery.

Molecules containing disulfide bonds are often selected to prepare colon delivery systems [89]. Qiao et al. developed a novel polymer by linking a hydrophilic molecule polyethylene glycol to a hydrophobic drug [90]. In vivo studies verified that the administration of the new polymer exerted a better therapeutic effect compared to sulfasalazine in murine colonic inflammatory model induced by dextran sulfate sodium [90].

Moreover, newly studied polysaccharides such as agave fructans, arabinoxylans, and other modified polisaccharides which are decomposable by colonic microbiota are applied to prepare targeted colon formulations [91,92].

##### ROS-Responsive Approach

Inflammation is involved in most intestinal diseases, reactive oxygen species (ROS) are usually produced during the process. ROS can overstep the antioxidant protection from physiological mechanisms, cause structural cell damage, and inflammation amplification.

High ROS levels lead to disfunctions of normal cell and cellular signaling transduction, promoting the inflammation process, which consequently aggravate the disease [93,94].

Huang Y. et al. [95] functionalized the surface of PLGA-based nanoparticles using pluronic F127 (PF127); catalase (CAT) and curcumin (CUR) were then encapsulated into the NPs.

The study found that loading of CAT could prominently increase the CUR release rate from NPs in H_2_O_2_-rich environment and the prepared NPs showed excellent capacity to suppress pro-inflammatory cytokines’ secretion.

Vong et al. developed oral redox nanoparticles that could specifically accumulate in inflamed colon and scavenge overproduced ROS, and thus suppress inflammation with low adverse effects [18,96]. Following the approach, a mimetic nanomedicine of superoxide dismutase/catalase was developed by Zhang et al.; the biocompatible nanomedicine was composed of a ß-cyclodextrin-derivative and loaded with tempol (Tpl/OxbCD NP) [94]. Tpl/OxbCD NP could notably concentrate to colon and significantly suppressed colites after oral administration. Zhang et al. developed 6-shogaol loaded PLGA/polylactic acid–polyethylene glycol–folate nanoparticles (PLGA/PLA-PEG-FA) [53]. The formulation could target inflamed tissue and reduce colitis symptoms and produce wound repairment in the ulcerative colitis mice.

##### pH-Dependent Approach

The colon exhibits a higher pH value, which could be used as a targeting approach for colon target delivery [48,97,98]. Naeem et al. reported nanoparticles encapsulating cyclosporine A with Eudragit FS30D and PLGA (E/PNPs) [99]. The combination of the two materials could reduce drug release in an acidic environment, and increase drug release at the colonic environment. Distribution studies showed E/PNPs could improve distribution of cyclosporine A to colon, ameliorate disease symptoms in the colitis mouse model, and exert an effective therapeutical effect.

Oshi et al. developed microcrystals encapsulating dexamethasone (DXMCs), with a coating by chitosan oligosaccharide, alginate, and Eudragit S 100 [100]. The coated DXMCs were pH-dependent, which showed a reduced initial burst drug release at low pH, and a sustained release at pH 7–4. In addition, the system was found to be more effective compared with other formulations in a mouse colitis model.

A celecoxib-loaded microparticles with Eudragit was prepared by Bazan et al. [101]. The study proved that the application of Eudragit S100 and Eudragit L100-55 could control the release of celocoxib, resulting in reducing colonic injury, colon inflammation, and inflammation markers.

##### Microbiota Dependent Drug Delivery Systems

Qiyan Chen et al. developed a hyaluronic acid, polyethylenimine, and modified rhein-loaded drug delivery system consisting of yeast cell wall microparticles (HA/PEI-RH NYPs) for ulcerative colitis (UC) treatment [102]. The yeast cell wall protected the microparticles from the harsh gastric environment and could be disintegrated by β-glucanase, produced by the microbiota in the colon, promoting drug release. In vivo studies proved that HA/PEI-RH NYPs could accumulate in the inflamed colon sites, inhibit the TLR4/MyD88/NF-κB pathway, and alleviate inflammation response. These results prove HA/PEI-RH NYPs to be a splendid oral drug delivery system for UC therapy.

### 3.2. Increased Bioavailability by Novel Drug Delivery System

Being susceptible in hostile gastrointestinal environment, drugs such as proteins and peptides had poor permeability across GIT mucosal and epithelial cells, resulting in inferior absorption and limited bioavailability by oral administration. The application of nanomedicine and microfabricated devices to address this issue have been supported by recent studies [14,103,104]. The main novel strategies used to increase bioavailability of oral drug delivery are shown in Table 2.

Therapeutic monoclonal antibodies are often used to treat gastrointestinal disease, while systemic injection of these agents may cause side effects, limiting the clinic use of these drugs. Oral delivery the therapeutical antibodies to GIT can be an effective approach for gastrointestinal diseases. A nanocomposite carrier was developed for infliximab to treat inflammatory bowel disease [105]. The infliximab can be delivered by nanocomposite carriers to colon and exert significant therapeutic effects, indicating the feasibility of oral delivery of antibodies for inflammatory bowel disease treatment.

Peptides and proteins are susceptive to physiological factors of GIT including stomach acid, peptidases, and pepsin, leading to limitations of oral administration. Studies have proven that peptides and proteins could survive from the hostile environment by encapsuling them in enteric-coated oral formulations [31,106,107]. This enteric-coated strategy is often used to control the diffusion of drugs, the properties of the coated membrane determine the rate of drug release, which obeys Ficks’ first law of diffusion.

A double coated nanocomposite system composed of organoclay, glycol-chitosan, and Eudragit^®^S100 was developed to enhance the absorption of insulin in colon by oral administration [108]. The nanocomplex demonstrated a pH-dependent drug release profile, enhanced drug permeability and absorption, and significantly reduced blood glucose levels in diabetic rats.

Pei et al. prepared polysaccharides coating calcium phosphate nanoparticles (CaP NPs) to delivery protein antigen orally [109]. An in vitro study revealed the enhancement of uptake by intestine epithelia cells and macrophages by CaP NPs. In vivo study further proved that nanoparticles significantly enhanced responses of mucosal IgA and serum IgG antibody after oral administration, indicating the nanoparticles can be used as a viable oral vaccine delivery system.

Chinnu Sabu et al. presented a promising bioinspired approach for insulin employing yeast microcapsule (YMC) [110]. The insulin-loaded yeast microcapsule (IYMC) coated with alginate was protected from the harsh environment and kept insulin stable in the microcapsule. The yeast component can be uptaken by M cells and absorbed into the systemic circulation via lymphatic transport. The study proved that the IYMC could exert an obvious hypoglycemic effect and have the potential to serve as an effective approach for insulin’s oral delivery.

Tianyang Ren et al. [111] reported an exenatide loaded yeast capsulated polylactide-co-glycoside nanoparticles (EXE-PLGA NPs @YCWPs) to protect EXE against gastrointestinal degradation and targeted delivery to phagocytic cell in intestine. EXE-PLGA NPs @YCWPs could efficiently reach the villi root in ileum and Peyer’s patches sites in biodistribution study and could be endocytosed by macrophages. EXE-PLGA NPs @YCWPs exerted an obvious hypoglycemic effect, improving the pharmacological availability after oral administration. Therefore, the PLGA NPs @YCWPs could serve as an efficient approach to orally deliver peptides.

An ingestible microfabricated device, termed the luminal unfolding microneedle injector (LUMI), was developed by Alex et al. [112]. LUMI was designed to consist of an elastomeric core and an enteric capsule shell. The elastomeric core was composed of three degradable arms, which were able to propel the drug loaded microneedle patch puncture into the tissue. When entering the intestine, LUMI could quickly unfold and expand within the intestine tract, resulting in increased retention and bioavailability in intestine. With the capacity to load a mass of microneedle formulations, the device can be applied as a platform to deliver therapeutic drugs by oral route.

Alex et al. developed an ingestible self-orienting millimeter-scale applicator (SOMA); it could be autonomously positioned and insert drug-loaded milliposts into the stomach lining, with an inspiration from the leopard tortoise’s ability to passively reorient [113]. The device was designed according to a geometric model of tortoise shell, it combined a low-density polycaprolactone and high-density stainless steel to form the self-orient low mass center. A 9-N steel spring (k = 1.13 N/mm) was used as a power source to release the drug tip through gastric mucosa without perforating the tissue. In vivo studies demonstrated that SOMA could increase plasma levels of active pharmaceutical ingredient.

Increasing the oral bioavailability of poorly water-soluble drugs has long been an elusive goal for oral drug delivery [114]. Chuang et al. developed a “Transformers”-like nanocarrier system (TLNS) to greatly enhance the oral bioavailability of curcumin (CUR), a poorly water-soluble drug [115]. The CUR-loaded TLNS could be absorbed and transferred by intestinal M cells, increase the intestinal drug dissolution, and notably increase the therapeutic efficacy of CUR to acute pancreatitis.

Naserifar et al. prepared resveratrol-loaded PLGA nanoparticles targeted with folate (PLGA-FA-RSV) to improve the stability of resveratrol and improve its pharmacokinetics and intestinal absorption. The study showed PLGA-FA-RSV could significantly enhance the transwell permeability rates, and significantly inhibit the inflammation and reduce accumulation of neutrophil and lymphocytes in colonic inflammation [116].

### 3.3. Targeting Delivery for Non-Gastrointestinal Diseases after Oral Administration

Different from intravenous administration, various new nano preparations can realize drug targeted delivery of various diseases through passive targeting brought by size and active targeting brought by targeting unit. The harsh environments of GIT have brought severe challenges to realize drug targeted delivery to non-gastrointestinal lesions, and the development has been very slow. With the discovery of the pathway and mechanism of pathogenic microorganisms invading the body through the GIT, the fate of these pathogenic microorganisms in vivo has provided pharmaceutical researchers a lot of enlightenment.

It is gradually possible to deliver drugs to the distal lesions of non-gastrointestinal diseases, the oral drug delivery systems are no longer limited in targeted drug delivery in the GIT. With the rapid progress of formulation technology and the in-depth understanding of disease pathophysiology, researchers have successfully achieved targeted administration to a variety of diseases such as systemic inflammation, tumor, brain diseases, cardiovascular diseases, obesity-related diseases, and arthritis through oral administration in recent years (Figure 2).

#### 3.3.1. Systematic Inflammation Target Delivery System

Due to the biological barriers, targeting delivery to systematic diseased sites through oral administration still remains challenging. On account of the ability to promote pathogenic inflammatory responses, macrophages are attractive targets in drug delivery for RNA interference. Myriam Aouadi et al. [117] developed siRNA particles (GeRPs) using β1,3-D-glucan, a yeast composition, to prepare efficient oral delivery systems. The particles can be phagocytized by macrophages and dendritic cells in the gut-associated lymphatic tissue (GALT) through the dectin-1 receptor and be transferred by GALT macrophages. GeRPs could deplete its messenger RNA in macrophages and decrease serum Tnf-α levels after oral administration. GeRPs could silence inflammation cytokine expression and prevent mice from lipopolysaccharide induced lethality.

Xing Zhou et al. [118] reported a targeted oral formulation based on yeast-derived capsule (YC) using a bioinspired ‘Trojan horse’ strategy; the design is shown in Figure 3. Polyethyleneimine (PEI), which can interact with carboxyl-containing guest molecules to form nanoparticles, was used as nanoparticle carrier. Nanoparticles containing a carboxylic anti-inflammatory drug indomethacin (IND) were packaged into YC. YCs are first transcytosed at Peyer’s patches by M cells after oral delivery, sequentially endocytosed and transported by macrophages to adjacent lymphocinesia, and ultimately delivered to inflammation sites through blood circulation. A study found that the inflamed rats treated with YCs had significantly higher levels of IND as compared to those of non-inflamed paws and other formulations treated groups after oral administration. Conclusively, this study proved that the yeast-derived bioinspired nanomedicine can be used for oral delivery to target systemic inflammation at diseased sites.

#### 3.3.2. Oral Tumor Target Delivery System

An abundance of tumor-targeting drug delivery agents has been developed in last decades, most of them being applied by the injection route. However, the development of a target delivery system for non-digestive tract tumors after oral administration still needs further work. The biomimetic oral approaches, such as the yeast-derived capsule (YC), have been proven to be highly promising in treating diverse diseases [19,118,119].

This yeast biomimetic system has also been proven to give oral application prospects to some drugs that could not be administered orally [120]. Cisplatin (CDDP) is an oral ineffective antitumor drug; however, Xing Zhou et al. [120] delivered it to a tumor by YCs orally. A nano precursor-packaging strategy was employed, CDDP was processed to a hydrosoluble prodrug and prepared into nanoparticles (Figure 4). The nanoparticles were then packaged into YC (PreCDDP/YC), showing significantly higher bioavailability compared to CDDP. PreCDDP/YC could accumulate to tumor sites in xenografts mice bearing A549 human lung carcinoma after oral administration. Moreover, oral PreCDDP/YC treatment exerted better safety than intravenous CDDP administration. This study demonstrated that the biomimetic system can be used to develop targeted oral chemotherapy for drugs such as CDDP or its derivatives.

Another biomimetic study was inspired by the spores; it generated an autonomous oral NP which can overcome the harsh GIT environment and diverse biological barriers [121].

Chemotherapeutic drugs, doxorubicin and sorafenib (DOX/SOR), were loaded in spores and modified with deoxycholic acid (Spore-DA); after entering the intestine, the DOX/SOR/Spore-DA germinated and self-assembled to NPs, the fabrication and transport mechanism of the autonomous nanoparticles is shown in Figure 5. The NPs were able to penetrate the epithelial cells through the bile acid pathway and increased basolateral drug release. Both in vitro and in vivo studies confirmed the formation of the autonomously produced NPs in biological nanogenerator, offering a promising approach for cancer drug delivery [121].

#### 3.3.3. Obesity-Related Diseases Target Delivery System

Obesity-related diseases are chronic diseases involving many diseases such as atherosclerosis, fatty liver, and diabetes. Due to its convenient and safe properties, oral administration is the best option for chronic disease treatment. Therefore, anti-obesity drugs are created based on oral administration, but it is still difficult to target drug delivery to lesions in obesity-related diseases (ORD) through oral administration. Fortunately, the recruitment of monocyte macrophages to ORD provides the possibility of oral obesity-targeting drug therapy.

Chunmei Xu et al. [122] presented a successful bindarit-loaded nanoparticles modified by laminarin (LApBIN) which could be used for obesity-related diseases. Their investigations in diet-induced obesity and atherosclerosis mouse models demonstrated that LApBIN could effectively target deliver drug to multiple lesions including inflammatory adipose tissue, fatty liver, and atherosclerosis; furthermore, LApBIN administrattion could also ameliorate insulin resistance (Figure 6). This oral targeted therapy strategy for various lesions provides an inspiring strategy to develop effective oral treatments for obesity and obesity-like systemic metabolic diseases.

Li Zhang et al. [123] also treated obese mice using yeast microcapsules to deliver IL-1b shRNA via the oral route. The oral shRNA/yeast microcapsules were prepared with non-virus-mediated interference vectors and non-pathogenic *Saccharomyces cerevisiae*. The oral treatment with IL-1b shRNA/yeast could notably reduce the body weight and fat weight of obesity mice and improve lipid metabolism-related cytokines and blood glucose concentration without diet-control.

#### 3.3.4. Gut-to-Brain Oral Drug Delivery System

The existence of intestinal epithelial barrier (IEB) and the blood–brain barrier (BBB) limits the drug delivery from gut to brain. Recently, a noninvasively prodrug approach was employed to overcome the IEB and BBB and deliver drugs to gliomas orally. The anticancer prodrug was prepared by conjugating with β-glucans. The prodrug could be targeted and absorbed by M cells and thus overcome IEB. Then, the transportation of the prodrug was dependent on macrophages, entered the circulatory system through lymphatic transport, and crossed the BBB. When the prodrug reached the tumor site, the overexpressed glutathione cleaved the prodrug and released the active drug. The design and transportation of the system is shown in Figure 7. The therapeutic efficacy of the drug was obviously improved by this gut-to-brain oral drug delivery platform, which might be excellent for glioma treatment [124].

#### 3.3.5. Cardiovascular Disease Oral Drug Delivery System

Many monocytes are related to cardiovascular diseases, such as atherosclerosis, which also provides a basis for achieving oral targeted drug delivery. A cardiovascular disease targeted treatment using biomimetic yeast-derived microcapsule (YC) approach was developed by Xiangjun Zhang et al. [125]. Positively charged nanodrugs are efficiently packaged into YC, and they can be uptaken by macrophages and maintained for a long time. The YCs were found to accumulate in aortic plaques in atherosclerotic apolipoprotein E-deficient (ApoE/) mice. Correspondingly, the YCs showed notably enhanced efficacy in atherosclerotic ApoE/mice. These results come to the conclusion that the YCs are just like a Trojan horse; they could be employed to orally deliver drugs to cure atherosclerosis and other vascular disorders.

#### 3.3.6. Post-Traumatic Osteoarthritis Oral Drug Delivery System

Gene therapies have been proven to improve diverse disease progressions, but they were limited due to lack of effective delivery systems. Long Zhang et al. [126] established a drug delivery system by non-pathogenic YC to orally delivery interleukin-1b (IL-1b) short hairpin RNA (shRNA) and evaluate its effect on post-traumatic osteoarthritis. The IL-1b shRNA loaded YC administration greatly reduced the expression of IL-1b in macrophages and diseased sites, resulting in significant inhibition of the inflammatory response systematically and articular cartilage degeneration of the knee joint. The expressions of osteoarthritis markers were also markedly lowered. The system proved that the YC-mediated oral delivery system might be a novel gene therapy strategy for treating osteoarthritis.

## 4. Conclusions

All the above studies collectively support the fact that the innovative pharmaceutical approaches can be applied to delivery therapeutical drugs to diverse diseases by oral administration. They can overcome multiple biological barriers and improve the effectiveness of various drugs. These promising results suggest that the oral drug delivery approaches can provide novel avenues for the treatment of other disorders, including neurodegenerative diseases, meningitis, and cellular injuries, for which effective approaches remain to be found.

However, current oral drug delivery systems focus more on improving the oral bioavailability of drugs and local targeting in the GIT. Although the targeting delivery to lesions beyond the GIT has been explored, it is still in its infancy; more targeting methods and strategies need to be developed. Most of the non-gastrointestinal targeting delivery systems are achieved based on the transportation by macrophages in vivo, but the specific transport route and mechanism are still unclear, which also needs to be further clarified in future studies. The targeting of systemic diseases based on macrophages also has some problems. The distribution and effect of macrophages are two-sided, when multiple lesions exist in the body simultaneously, the transportation of macrophages may lead to incorrect targeting sites. This issue may lead to side effects which need to be settled in future work.

Meanwhile, most of the existing oral targeted drug delivery systems can deliver drugs to organs and tissues; with the development of drug delivery technology, the future systems can achieve results of targeting to cells and organelles after oral administration, which will improve the efficiency of targeting. In addition, these innovative delivery systems such as nanomedicines are still facing translational challenges that limit the practical application of drugs in clinical practice. In contrast to the fact that most of the drugs are produced for oral application, the overwhelming majority of the FDA-approved nanomedicines are administrated by intravenous injection worldwide [4]. Currently, the widespread application of nanomedicine therapeutics is hampered due to the nanomaterial characterization, safety concerns, and manufacturing issues.

Furthermore, normative criteria and guidelines for both regulatory approval and industrial production are urgently needed [127]. Once these problems are solved, the conventional injection formulations could be replaced by the novel oral drugs, bringing more convenience to millions of people and offering hitherto undescribed possibilities.

## Figures and Tables

**Figure 1 pharmaceutics-15-00484-f001:**
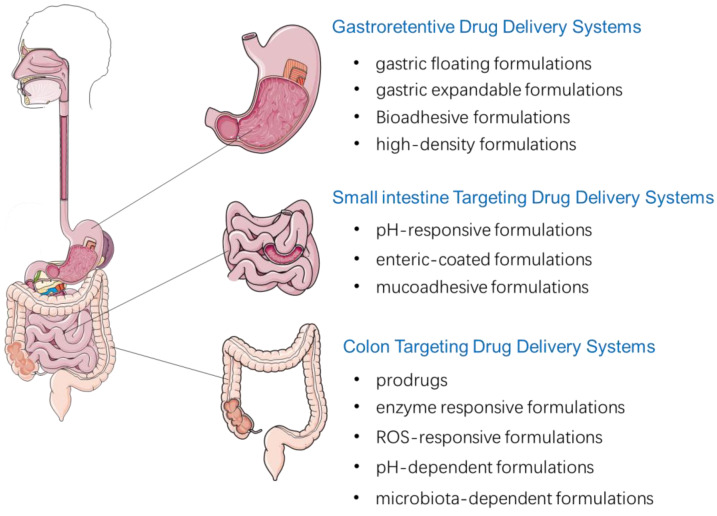
Local targeting drug delivery systems to GIT after oral administration.

**Figure 2 pharmaceutics-15-00484-f002:**
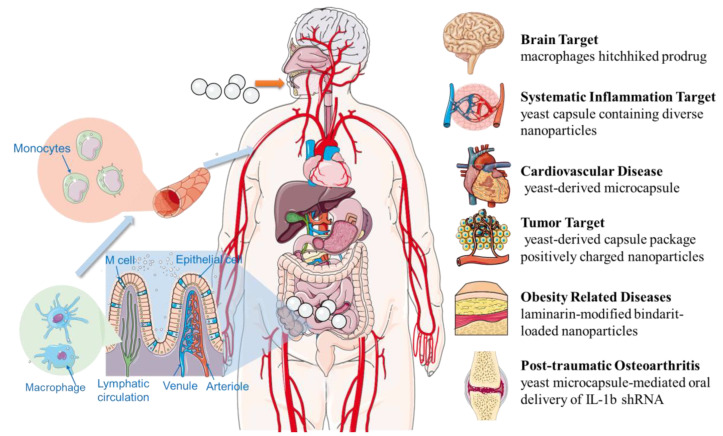
Targeting delivery systems to non-gastrointestinal diseases after oral administration.

**Figure 3 pharmaceutics-15-00484-f003:**
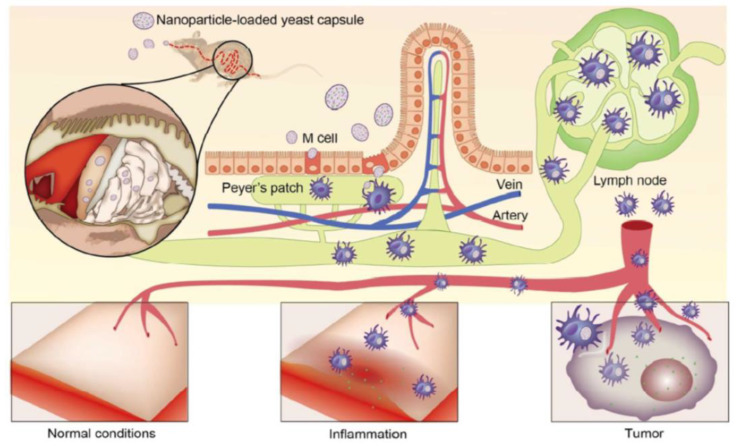
Schematic diagram of YC-mediated nanoparticles to inflammation-associated diseases [118]. Reproduced from ref. [118] with permission from the American Chemical Society, copyright 2017.

**Figure 4 pharmaceutics-15-00484-f004:**
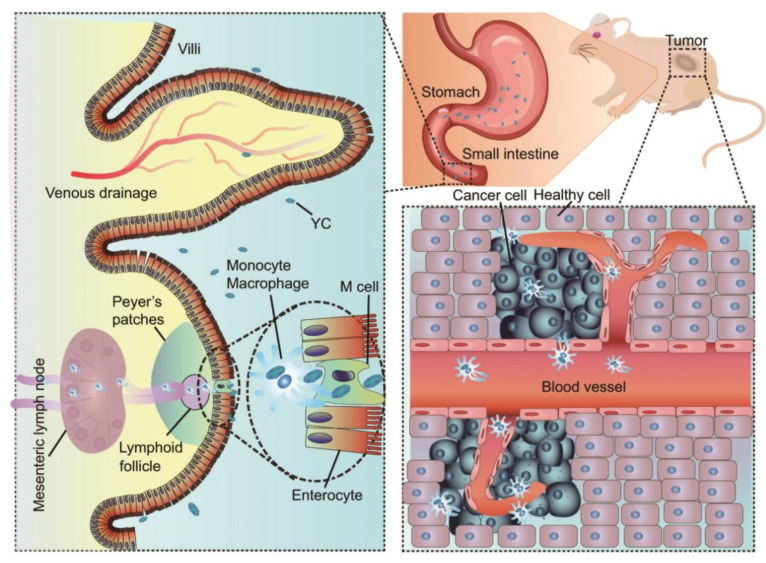
Schematic diagram of a CDDP nanoprecursor for targeted tumor treatment [120]. Reproduced from ref. [120] with permission from the author.

**Figure 5 pharmaceutics-15-00484-f005:**
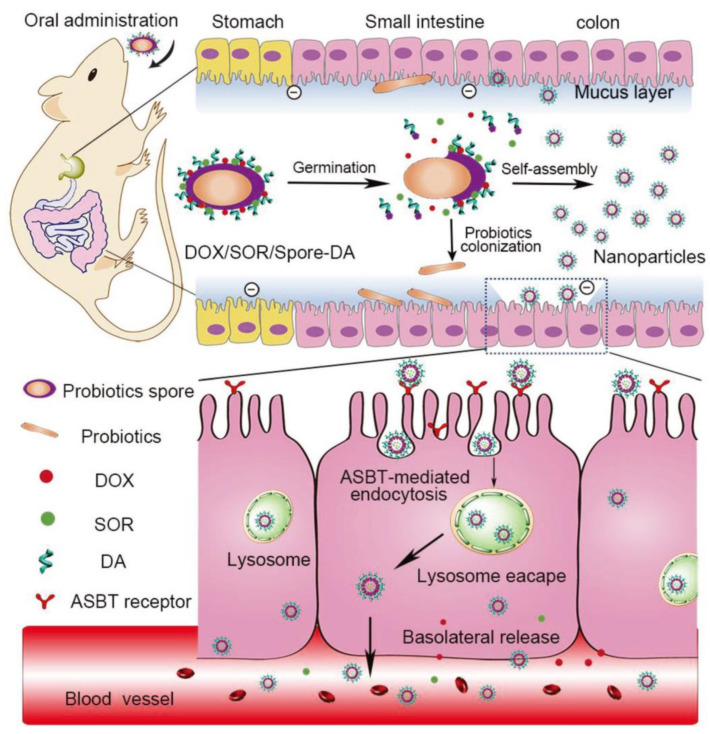
Schematic illustration of the fabrication and transepithelial transport mechanism of DOX/SOR/Spore-DA [121]. Reproduced from ref. [121] with permission from the John Wiley and Sons, copyright 2018.

**Figure 6 pharmaceutics-15-00484-f006:**
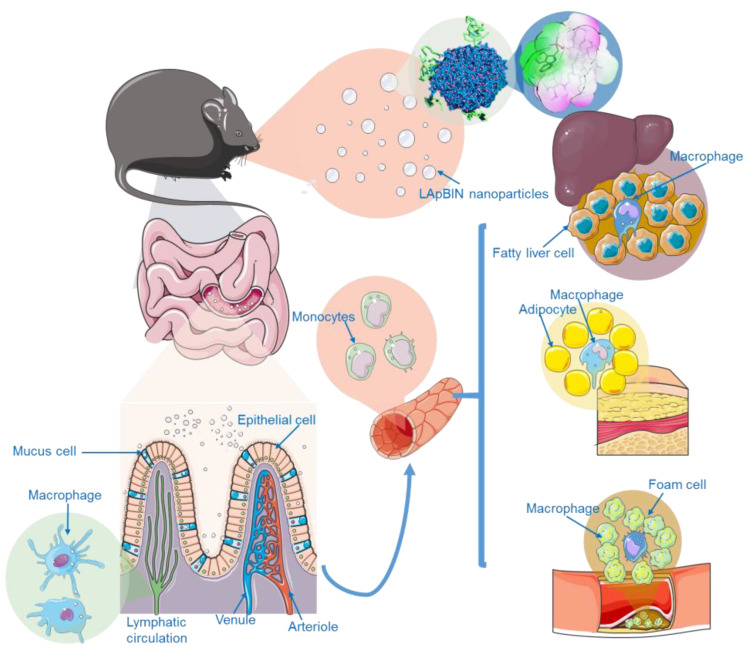
Schematic diagram of LAM-mediated oral targeting of nanoparticles to multiple sites of obesity related diseases [122]. Reproduced from ref. [122] with permission from the author.

**Figure 7 pharmaceutics-15-00484-f007:**
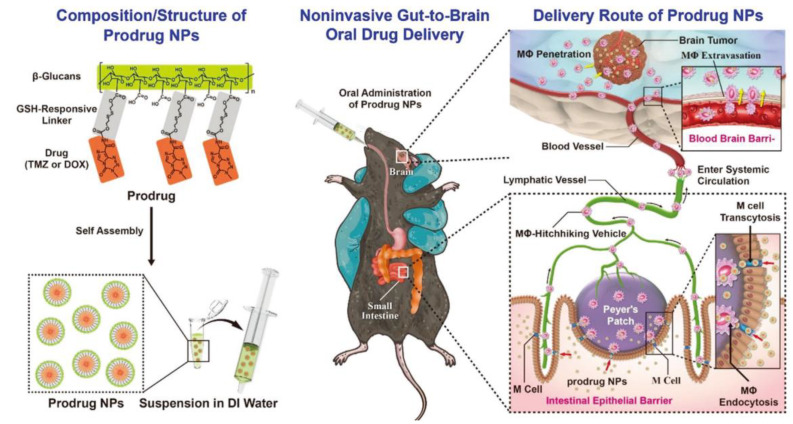
Schematic diagram of the noninvasive gut-to-brain oral drug delivery system [124]. Reproduced from ref. [124] with permission from John Wiley and Sons, copyright 2021.

**Table 1 pharmaceutics-15-00484-t001:** Overview of main anatomical and biochemical barriers of the oral administration [21,22,23,24,25,26,27,28,29,30,31,32,33].

Site of Gastrointestinal Tract	Anatomical Factors	Biochemical Barriers			References
		pH Values	Enzyme	Microbiome	
Mouth cavity	limited surface, saliva, enzymatic	6.6–7.1	Salivary amylase	Gemella, Streptococcus, Veillonella, Haemophilus, Neisseria, Porphyromonas, Actinomyces, Fusobacterium, Prevotella	[20,22,37]
Stomach	gastric acid, mucin-bicarbonate barrier, pepsin, limited absorption area	1.5–2.2	Pepsin Amylase	Prevotella, Streptococcus, Veillonella, Rothia, Haemophilus	[25,28,37]
Small intestine	pancreatic enzymes, bile salts, mucosal layer	5.9–7.8	Pancreatic amylase, Maltase, Trypsin, Peptidases, Aminopeptidase, Nuclease, Nucleosidases, Lipase	Clostridiales, Streptococcus, Bacteroides, Actinomyxinae, Lactobacillus, Corynebacteria, Enterrococcus	[31,32,37]
Colon	gut microflora, enzymes produced by microflora	6–6.7	Aminopepti-dase, Azoreductase, Polysaccharase, Glycosidase, Nitroreductase	Porphyromonas, Eubacterium, Bacteroides, Bifidobacterium, Streptococcus, Enterobacteriaceae, Enterococcus, Clostridium, Lactobacillus, Ruminococcus, Escherichia coli	[33,34,37]

**Table 2 pharmaceutics-15-00484-t002:** Increased bioavailability by novel drug delivery system.

Formulation Strategy	Drug	Graph of the Formulation	Reference
Nanocomposite carriers	infliximab	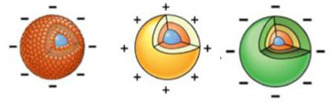	[105]
Coated nanocomplex	insulin	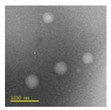	[108]
Calcium phosphate nanoparticles coated with polysaccharides	BSA or OVA	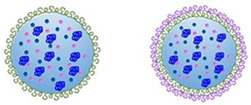	[109]
Yeast microcapsule	insulin	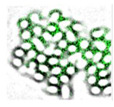	[110]
PLGA nanoparticles in yeast cell wall particle	exenatide	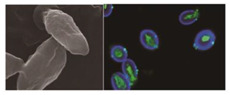	[111]
Luminal unfolding microneedle injector	insulin	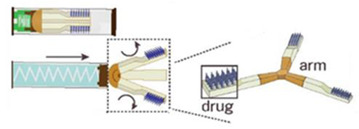	[112]
Ingestible self-orienting millimeter-scale applicator	insulin	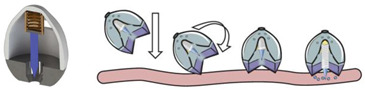	[113]

## Data Availability

Not applicable.

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
