# Peer review of "Advances in Oral Drug Delivery Systems: Challenges and Opportunities"

_pharmaceutics, 2023, doi:10.3390/pharmaceutics15020484_

Round 1
Reviewer 1 Report
The manuscript structure and content are good but there are many mistakes and typo errors present. This manuscript can be recommended for publication after minor revision.
Here are some comments that need to be addressed as below:
1. The complete manuscript needs to be rechecked and proofread before publication to avoid typos and grammatical errors. I found many typo errors in line no. 52, 181, 187, 204, 208, 227, 232, etc.
2. All the tables and figures should be cited in the text. I found tables 1 and 2, some figures are also not cited in the text.
3. The typo errors were also found in the references cited in the text. Please go through it and corrections should be done.
4. In table 1 references should be added like in table 2.
5. In line no. 232 the polymer name should be corrected.
Author Response
Response to Reviewer 1 Comments:
Comments and Suggestions for Authors: The manuscript structure and content are good but there are many mistakes and typo errors present. This manuscript can be recommended for publication after minor revision.
Response: We would like to thank the reviewer for these positive comments and good suggestions. With your suggestion, we have significantly improved the quality of this manuscript.
Point 1: The complete manuscript needs to be rechecked and proofread before publication to avoid typos and grammatical errors. I found many typo errors in line no. 52, 181, 187, 204, 208, 227, 232, etc.
Response 1: Thanks for your comments which are really useful to improve our manuscript. We have carefully checked and corrected the typos and grammatical errors in the revised manuscript.
Point 2: All the tables and figures should be cited in the text. I found tables 1 and 2, some figures are also not cited in the text.
Response 2: Thanks for your comments, we have made revisions and the tables and figures were cited in the revised manuscript.
Point 3: The typo errors were also found in the references cited in the text. Please go through it and corrections should be done.
Response 3: Thanks for your comments, we have carefully checked and corrected errors in the revised manuscript.
Point 4: In table 1 references should be added like in table 2.
Response 3: Thanks for your suggestion, we have added references in table 2.
Point 5: In line no. 232 the polymer name should be corrected.
Response 3: Thanks for your comments, we have corrected the polymer name in the revised manuscript.

Reviewer 2 Report
The article is well-written and well-stuctured.
I do not want to modify. I accept in the present form.
Author Response
Comments and Suggestions for Authors: The article is well-written and well-stuctured. I do not want to modify. I accept in the present form.
Response: We sincerely thank the reviewers for your careful review and comments. We are really glad to receive your approval. All the best to you.

Author Response
Comments and Suggestions for Authors: The review manuscript entitled “Advances in Oral Drug Delivery Systems: Challenges and Opportunities” presented a review on various biological factors that affect the oral administration and applications of nanomedicines in and beyond GI tract. This review is a well-organized and fits scope of Pharmaceutics. I have few suggestions given below.
Response: Thanks for your positive comments. According to your suggestion, we have significantly improved the quality of this manuscript.
Point 1: There are some typos in manuscript; line 70 on page 2 ‘nanomedcine’ -> ‘nanomedicine’, line 265 on page 7 = ‘pecific’ -> ‘specific’
Response 1: Thanks for your careful review and helpful comments. We have carefully checked and corrected the typos in the revised manuscript.
Point 2: For Figure 1 and 2, there are no “Figure 1” and “Figure 2” marks in the manuscript. It would be better if the “Figure 1” and “Figure 2” are added to each relevant sentence in the manuscript.
Response 2: Thanks for your careful review, we have made revisions and the figures were added to relevant contents in the revised manuscript.
Point 3: For “2. Application of the oral drug delivery systems” section, the publication date of most references is a bit old. I would suggest the authors to consider citing one or two latest references to each section.
Response 3: Thanks for your suggestion, we have carefully studied the recent studies and updated latest references in this section.
Point 4: It would be interesting if the authors briefly describe perspectives, challenges, and future research directions.
Response 4: We sincerely appreciate for your suggestion. We briefly summarized the current status and limitations of oral drug delivery systems, and the future research directions of oral drugs, which we hope would make this manuscript more sophisticated.

Reviewer 4 Report
The manuscript is an extensive review that presents a very interesting theme: oral drug delivery systems; it contains a huge amount of data (125 references, most of them are very recent) about various drugs, bioactive agents and many carriers. However, the following remarks are essentials:
(1) please add a few remarks related to the modelling of diffusion controlled drug delivery and the Fick's laws of difussion
(2) please use italics for: et al, in vivo, in vitro etc.
(3) it is necessary to verify the text in order to remove small English grammar and typo errors
(4) in Table 2: you must change the name of the column "Schematic diagram" because not all images are schemes
Author Response
Comments and Suggestions for Authors:The manuscript is an extensive review that presents a very interesting theme: oral drug delivery systems; it contains a huge amount of data (125 references, most of them are very recent) about various drugs, bioactive agents and many carriers. However, the following remarks are essentials:
Response: Thanks a lot for your positive comments and valuable suggestions. With your suggestion, we have significantly improved the quality of this manuscript.
Point 1: please add a few remarks related to the modelling of diffusion controlled drug delivery and the Fick's laws of difussion.
Response 1: Thanks for your suggestions, we have added related contents about the enteric-coated strategy in the revised manuscript.
Point 2: please use italics for: et al, in vivo, in vitro etc.
Response 2: Thanks for your suggestions, we have made revisions in the revised manuscript.
Point 3: it is necessary to verify the text in order to remove small English grammar and typo errors
Response 3: Thanks for your comments. We have carefully checked and corrected the typos and grammatical errors in the revised manuscript.
Point 4: in Table 2: you must change the name of the column "Schematic diagram" because not all images are schemes
Response 4: Thanks for your comments, we have changed the "Schematic diagram" into “Graph of the formulation” in Table 2, which we believe would be more proper.
